# Selection of chromosomal DNA libraries using a multiplex CRISPR system

Owen W Ryan[1][†], Jeffrey M Skerker[1,2,3], Matthew J Maurer[1], Xin Li[1], Jordan C Tsai[1,4], Snigdha Poddar[1], Michael E Lee[1,2], Will DeLoache[1,2], John E Dueber[1,2], Adam P Arkin[1,2,3], Jamie HD Cate[1,3,4,5]*

[1]Energy Biosciences Institute, University of California, Berkeley, Berkeley, United States; [2]Department of Bioengineering, University of California, Berkeley, Berkeley, United States; [3]Physical Biosciences Division, Lawrence Berkeley National Laboratory, Berkeley, United States; [4]Department of Molecular and Cell Biology, University of California, Berkeley, Berkeley, United States; [5]Department of Chemistry, University of California, Berkeley, Berkeley, United States

**Abstract** The directed evolution of biomolecules to improve or change their activity is central to many engineering and synthetic biology efforts. However, selecting improved variants from gene libraries in living cells requires plasmid expression systems that suffer from variable copy number effects, or the use of complex marker-dependent chromosomal integration strategies. We developed quantitative gene assembly and DNA library insertion into the *Saccharomyces cerevisiae* genome by optimizing an efficient single-step and marker-free genome editing system using CRISPR-Cas9. With this Multiplex CRISPR (CRISPRm) system, we selected an improved cellobiose utilization pathway in diploid yeast in a single round of mutagenesis and selection, which increased cellobiose fermentation rates by over 10-fold. Mutations recovered in the best cellodextrin transporters reveal synergy between substrate binding and transporter dynamics, and demonstrate the power of CRISPRm to accelerate selection experiments and discoveries of the molecular determinants that enhance biomolecule function.

*For correspondence:
jcate@lbl.gov

Present address: [†]BP Biofuels Global Technology Center, San Diego, United States

## Introduction

Directed evolution using living systems allows selections for improved biomolecule function to be directly coupled to phenotype. However, present directed evolution systems require the use of extra-chromosomal gene libraries encoded in bacteriophage or plasmids, which suffer from high levels of copy number variation (*Yokobayashi et al., 2002*; *Turner, 2009*; *Esvelt et al., 2011*). For example, in the yeast *Saccharomyces cerevisiae*, common yeast centromere containing (CEN) plasmids vary widely in copy number per cell resulting in highly variable expression levels from cell to cell (*Figure 1*). Ideally, the selective pressure applied to evolve improved or new biomolecule function should be limited to the DNA sequence level and not to gene copy number in a given cell (*Zhou et al., 2012*). To overcome copy number variation in yeast, the standard genetic tool of integrating linear DNA into the genome by homologous recombination is too inefficient to insert DNA libraries without multiple steps that rely on dominant selectable markers for chromosomal integrations (*Wingler and Cornish, 2011*) or specialized strain backgrounds (*DiCarlo et al., 2013a*).

Bacterial type II CRISPR-Cas9 genome editing has been used successfully in several eukaryotic organisms (*Cong et al., 2013*; *DiCarlo et al., 2013b*; *Mali et al., 2013a, 2013b*; *Shalem et al., 2014*) but has not been adapted for selection experiments. Genome editing CRISPR-Cas systems require a Cas9 endonuclease that is targeted to specific DNA sequences by a non-coding single guide RNA (sgRNA) (*Jinek, et al., 2012*; *Jinek et al., 2014*). The Cas9-sgRNA ribonucleoprotein complex

**eLife digest** Over the course of billions of years, natural evolution has produced new proteins and adapted existing ones so that they work better. Scientists have learned how to use the principles that underlie evolution to similarly engineer proteins in the laboratory. This process, known as directed evolution, is a powerful tool for improving how proteins function. Directed evolution normally involves mutating the gene that encodes the protein of interest, selecting the genes that produce the most promising proteins for another round of mutation, and repeating the process until the desired protein function is achieved.

In the first step of directed evolution, a gene is usually mutated randomly in order to create a large 'library' of different forms of the gene. These are joined to circular pieces of DNA known as plasmids that can replicate themselves inside cells. However, the number of plasmids than can be taken up differs from cell to cell. This complicates experiments, and the ideal directed evolution experiment would have the same number of plasmids, or target genes, being delivered into each cell.

Ryan et al. have developed a new method for performing directed evolution experiments that uses a recently developed technique called the CRISPR-Cas9 system. This can make direct changes to a DNA strand such as inserting or deleting specific sequences that code for proteins. Ryan et al. used the CRISPR-Cas9 system to create multiple DNA breaks simultaneously across the genome of yeast cells, and joined 'barcoded' DNA or DNA for intact genes to these breaks. This avoids the need to use plasmids to introduce foreign DNA into cells. Ryan et al. have named this method the Multiplex CRISPR (or CRISPRm) system.

Having established CRISPRm, Ryan et al. tested whether it could be used to engineer improved proteins by attempting to modify a transporter protein called CDT-1. This protein transports the sugar cellobiose into yeast cells, where it can be converted into alcohol by fermentation. This is important for making biofuel from plants. After just one round of directed evolution using CRISPRm, Ryan et al. successfully isolated a form of the CDT-1 protein that increased the rate of fermentation over 10-fold; hence this CDT-1 variant could be used to increase biofuel production.

In the future, it will be important to implement multiple selection rounds with CRISPRm, and to test how large the DNA libraries can be for directed evolution. In time, CRISPRm could find use in evolving and engineering different combinations of genes, metabolic pathways, and possibly entire genomes.

precisely generates double-strand breaks (DSBs) in eukaryotic genomes at sites specified by a 20-nucleotide guide sequence at the 5' end of the sgRNA that base pairs with the protospacer DNA sequence preceding a genomic protospacer adjacent motif (PAM) (*Sternberg et al., 2014*). The presence of the Cas9-produced DSB in genomic DNA can increase the rate of homology-directed repair (HDR) with linear DNA at the DSB locus by several 1000-fold (*Storici et al., 2003*; *DiCarlo et al., 2013b*) potentially enabling high-throughput experimental methods. CRISPR-Cas9 could therefore be useful in probing industrial *S. cerevisiae* strains, which are more robust compared to laboratory strains (*Kerr and Service, 2005*; *Farrell et al., 2006*; *Rubin, 2008*), but for which few genetic tools are available, due to the fact that these strains are often polyploid with low-efficiency mating and sporulation.

## Results

To optimize the efficiency of HDR in *S. cerevisiae* mediated by Cas9, we first constructed a plasmid expressing *Streptococcus pyogenes* Cas9 fused to a nuclear localization signal (Cas9-NLS) that efficiently localizes to the nucleus of yeast cells (*Figure 2A,B*; *Jinek et al., 2012*). We used an intermediate strength promoter, $P_{RNR2}$, to express Cas9 for genome targeting experiments, because Cas9 (Cas9-NLS-His8x) expression from the $P_{RNR2}$ promoter resulted in yeast strains with near wild type fitness whereas Cas9 expressed using strong yeast promoters such as $P_{TDH3}$ (*Lee et al., 2013*) reduced yeast fitness relative to wild type cells (*Figure 2C*). Second, we designed a new sgRNA architecture in order to improve its expression and function. Cellular levels of sgRNAs correlate with the efficiency of Cas9-mediated genome targeting in mammalian cells (*Hsu et al., 2013*). Controlled expression of nuclear-localized RNAs and the correlation between sgRNA levels and Cas9-editing in yeast has not

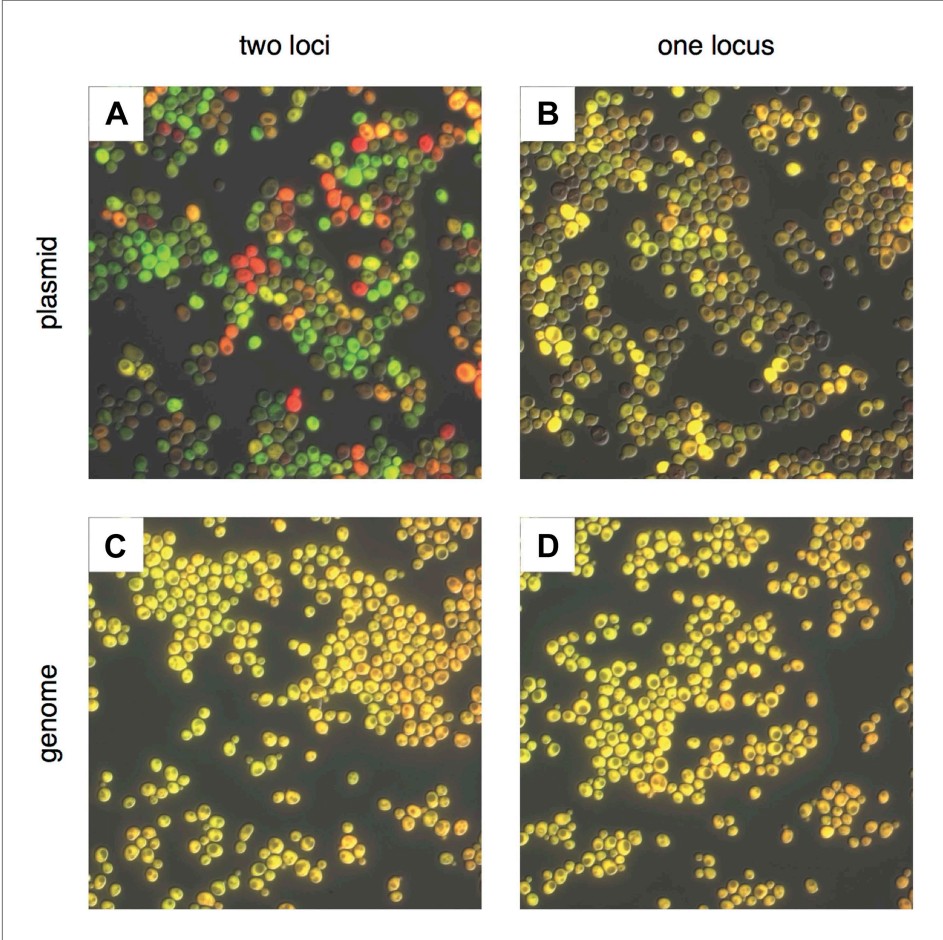

**Figure 1**. Comparing noise in plasmid and genomic expression of two fluorescent proteins. Proteins mRuby2 (false-colored red) and Venus (false-colored green) were expressed in the same cell with a strong, $P_{TDH3}$ promoter. (**A**) Fluorescent proteins expressed on two separate CEN plasmids; (**B**) in tandem on a single CEN plasmid; (**C**) integrated into two different loci (*LEU2* and *URA3*) in the *S. cerevisiae* genome; and (**D**) integrated in tandem at a single locus (*URA3*) in the genome. Each image was auto-exposed for both red and green channels, with yellow showing co-expression of mRuby2 and Venus. The variability in the relative expression of the two fluorescent proteins is reduced by moving from two plasmids to one, and the variability in total expression is reduced by moving to the genome. The difference between integrating into one or two loci in the genome is minimal.

been explored. To better control cellular levels of correctly folded sgRNA, we developed a modular design by fusing the sgRNA (*Mali et al., 2013a*) to the 3′ end of the self-cleaving hepatitis delta virus (HDV) ribozyme (*Ke et al., 2007*; *Webb et al., 2009*; *Figure 2D*). We reasoned that the structured ribozyme would protect the 5′ end of the sgRNA from 5′ exonucleases (*Houseley and Tollervey, 2009*). Additionally, the HDV ribozyme would cleave the RNA immediately 5′ of the ribozyme, removing extraneous RNA sequences and allowing flexibility in which promoters can be used, such as tRNAs whose DNA sequences also serve as RNA polymerase III promoters (*Marck et al., 2006*). Ribozyme removal of the tRNA would then separate transcription initiation from within the tRNA sequence (*Marck et al., 2006*) and tRNA nuclear export (*Köhler and Hurt, 2007*) from the process of forming functional sgRNA complexes with Cas9.

To quantify the efficiency and specificity of HDR with the new Cas9/sgRNA format, phenotype screens were performed by the co-transformation of a single plasmid that expresses the Cas9 protein and one or more HDV ribozyme-sgRNAs, containing a 20-nucleotide guide sequence that matches a specific site in the yeast genome, along with a linear double-stranded DNA (*Figure 2E*). The linear dsDNA, which contains a unique 20-mer barcode (*Giaever et al., 2002*) flanked by common primer sequences and 50 base pairs of DNA homologous to the regions proximal to the PAM motif

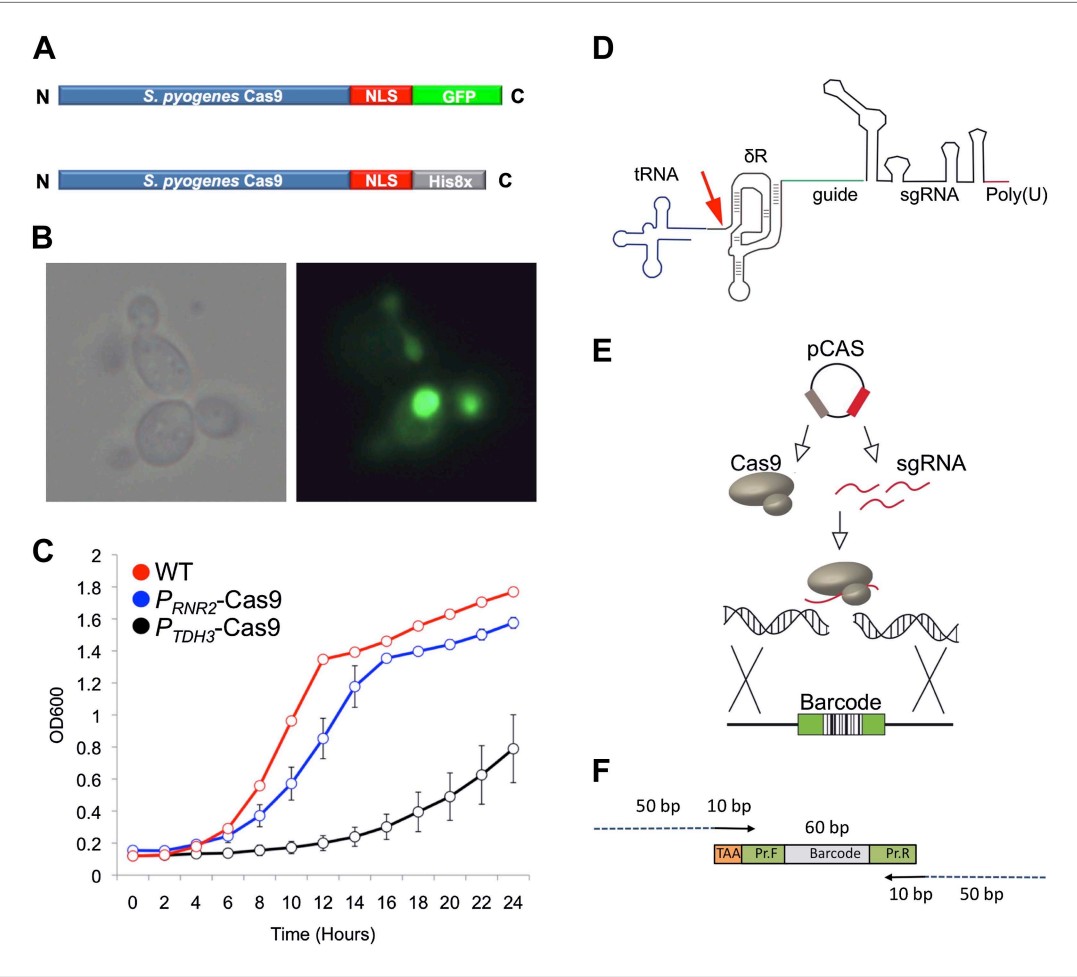

**Figure 2**. The engineered CRISPRm platform. (**A**) Cas9 construct used for nuclear localization experiments. The *S. pyogenes* Cas9 protein was tagged with a C-terminal nuclear localization motif and a green fluorescent protein (GFP) (green). For genome editing experiments, the *S. pyogenes* Cas9 protein was tagged with a C-terminal nuclear localization motif and a Histidine affinity tag (gray). (**B**) Cellular localization of Cas9-GFP in exponentially growing *S. cerevisiae* cells. The Cas9-GFP protein was expressed from the *TDH3* promoter in this experiment. Left, bright field image; right fluorescence microscopy. (**C**) Growth profiles of yeast expressing Cas9 from a strong promoter $P_{TDH3}$ (black) or a weaker promoter $P_{RNR2}$ (blue) relative to wild type (red). (**D**) The mature sgRNA contains a 5' hepatitis delta virus (HDV) ribozyme (δR, brown), 20mer target sequence (green), sgRNA (black) and RNA polymerase III terminator (red). The RNA polymerase III promoter tRNA (blue) is catalytically removed by the HDV ribozyme (red arrow). (**E**) Schematic of yeast Cas9 targeting. Cas9 and sgRNA are expressed from a single plasmid, form a complex, and cleave targeted genomic DNA, which is repaired using a barcoded oligonucleotide. (**F**) The linear barcoded repair DNA molecule. Each repair DNA contains a 5' TAA stop codon (gold), a forward primer sequence (green), a unique molecular barcode (gray), and a 3' reverse primer (green) (*Giaever et al., 2002*). Barcodes are amplified using a forward primer that contains 50 bp of homology (blue) to the genome targeting site and a reverse primer that contains 50 bp of homology to the genome targeting site. The 50 bp of genomic targeting sequence are each 10 bp proximal to the PAM motif, resulting in a 20 nt deletion and barcode oligonucleotide integration.

(*Figure 2F*), acts as a template for DNA repair by HDR, resulting in a unique markerless and barcoded insertion allele. With a single sgRNA, we found tRNA sequences used as RNA polymerase III promoters resulted in nearly 100% efficient barcode insertion in diploid yeast at the *URA3* locus, resulting in 5-fluororotic acid resistance (*Boeke et al., 1984*) whereas the non-tRNA promoters were mostly ineffective (*Figure 3A*; *Supplementary file 1A*). We also assessed target sequence bias by targeting 11 unlinked yeast genes in diploid S288C yeast cells using tRNA^Tyr as the RNA polymerase III promoter

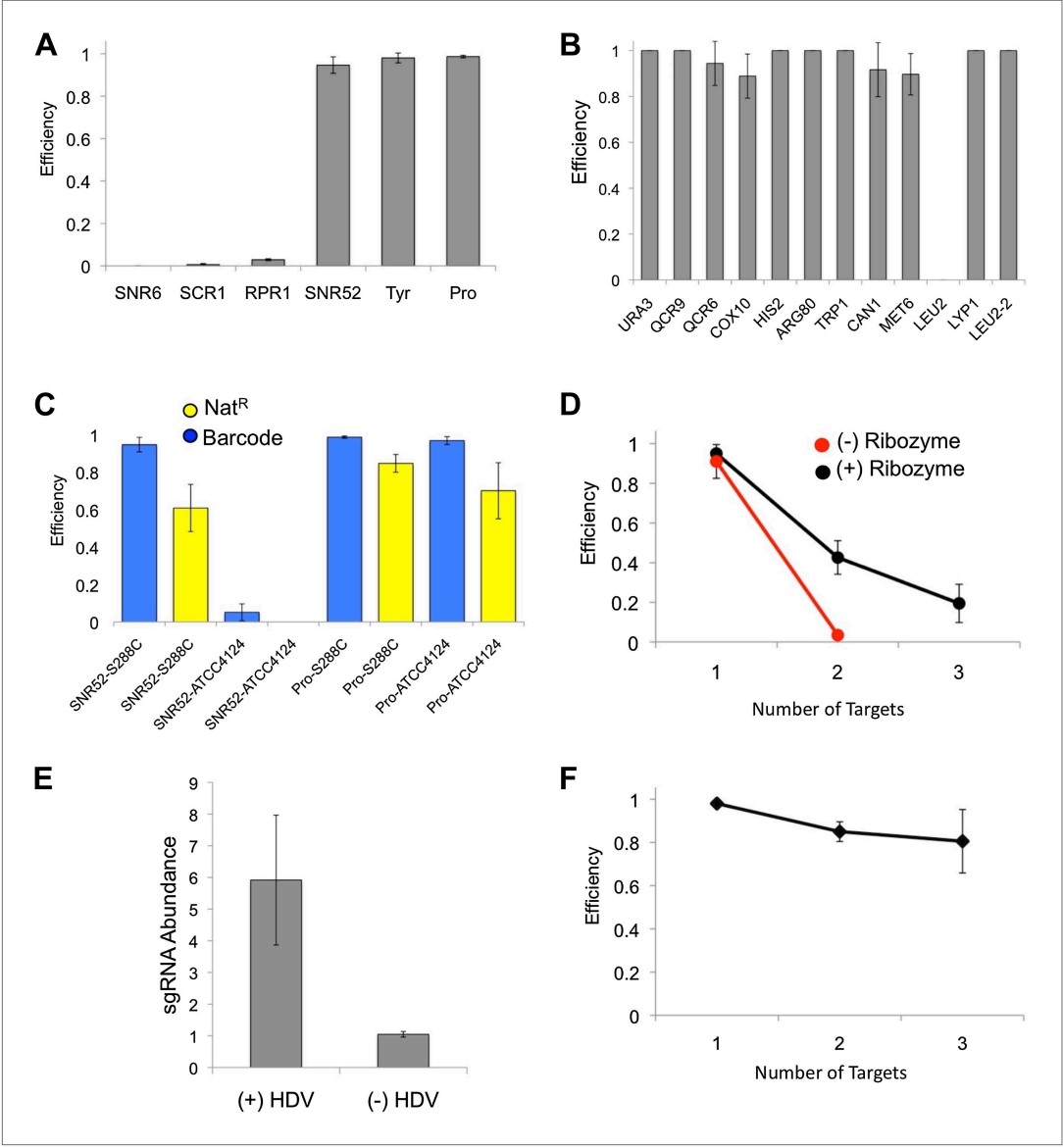

**Figure 3**. CRISPRm barcode insertion in yeast cells. (**A**) Targeting efficiency at the *URA3* locus in diploid S288C yeast using RNA polymerase III promoters (x axis) to drive the expression of the sgRNA. (**B**) Targeting efficiency measured across 11 loci in 10 genes in diploid cells of yeast strain S288C, using the tRNA$^{Tyr}$ promoter. (**C**) Single fragment barcode integration (blue) and three-fragment Nat$^R$ cassette integration (yellow) efficiency in S288C diploid and ATCC4124 polyploid strains. For each experiment the promoter and strain are indicated as promoter-strain (e.g., SNR52-S288C). (**D**) Efficiency of multiplex insertion of barcoded DNA in diploid yeast cells with the 5' HDV ribozyme (black) and without the 5' HDV ribozyme (red). Triplex targeting without the 5' HDV ribozyme was not tested. The tRNA$^{Tyr}$ promoter was used in these experiments. (**E**) The addition of a 5' HDV ribozyme increases the intracellular levels of sgRNA by sixfold. The RT-qPCR experiments were carried out in biological triplicate, with the mean and standard deviation shown. (**F**) Efficiency of targeting one (*URA3*), two (*URA3, LYP1*) and three (*URA3, LYP1* and *COX10*) in haploid S288C yeast cells. The tRNA$^{Tyr}$ promoter was used for sgRNA expression.

(*Supplementary file 2A*). We found that targeting was highly efficient for every genomic target except *LEU2,* which at first was weak but was restored to 100% by selecting a different guide sequence (*LEU2-2*) as the sgRNA (*Figure 3B*). We also verified that our CRISPR-Cas9/sgRNA system did not result in non-specific genome targeting by sequencing 9 of the targeted strains (*Supplementary file 1B*). We also tested efficacy of our approach in the polyploid industrial *S. cerevisiae* strain ATCC4124 that has superior tolerance and productivity phenotypes (*Ness et al., 1993*). The non-tRNA promoter

previously used in haploid yeast (*DiCarlo et al., 2013b*), $P_{SNR52}$, functioned in diploid S288C yeast but failed to result in targeting in the polyploid industrial yeast ATCC4124; in contrast, the tRNA$^{Pro}$ promoter enabled highly-efficient barcode insertion into all copies of the *URA3* locus in ATCC4124, which we term *cis*-multiplexing, that is introducing a double-stranded DNA break at a single genomic locus across all chromosomes (*Figure 3C*; *Supplementary file 1A*). Notably, tRNA$^{Tyr}$ was inefficient at targeting in ATCC4124 though it was effective in S288C diploid (*Supplementary file 1A*), indicating that the choice of tRNA used for the expression of the HDV-sgRNA impacts multiplexing in a strain-dependent manner.

Since the efficiency of *cis*-multiplexing with our CRISPR-Cas9/sgRNA system is nearly 100% using one sgRNA, we next tested its ability for *trans*-multiplexed CRISPR-Cas9 genome editing, that is introducing double-stranded DNA breaks at multiple genomic loci simultaneously, using tRNAs to drive the expression of the HDV-sgRNAs. We simultaneously targeted two and three unlinked genomic loci in diploid cells for loss-of-function mutagenesis by introducing a single plasmid expressing Cas9 and two or three sgRNAs with two or three gene-specific barcoded DNA molecules, respectively. The efficiency of duplex and triplex targeting (requiring four and six chromosome cuts, respectively) with two sgRNAs (*URA3* and *LYP1*) or three (*URA3*, *LYP1* and *COX10*) was 43% and 19%, respectively (*Figure 3D*; *Supplementary file 1A*). This multiplexed efficiency was dependent on the presence of the HDV ribozyme 5′ of the guide sequence as duplex targeting dropped to 3.5% in cells with sgRNA lacking the 5′ HDV ribozyme (*Figure 3D*; *Supplementary file 1A*). We measured the relative cellular abundance of sgRNAs expressed by $P_{SNR52}$, with and without the 5′ HDV ribozyme, by reverse transcription quantitative polymerase chain reaction (RT-qPCR) and found that the presence of the ribozyme increases the intracellular abundance of the sgRNAs by sixfold (*Figure 3E*), consistent with the model that sgRNA abundance is rate limiting for CRISPR/Cas targeting (*Hsu et al., 2013*). Haploid *trans*-multiplexing showed a minimal loss in activity scaling from one to three loci (up to three chromosome cuts) (*Figure 3F*; *Supplementary file 1A*). These experiments demonstrate that our multiplexed CRISPR/Cas9 system, which we term CRISPRm, is powerful enough to generate multiple marker-free targeted mutations in the yeast genome in a single experiment and that the 5′ tRNA-HDV ribozyme sequence is required for higher order multiplex targeting.

Having established CRISPRm for *cis*- and *trans*-multiplex targeting, we next tested its capabilities in engineering genes and pathways integrated in the yeast genome. We first tested whether CRISPRm could be used to assemble foreign genes as chromosomal integrations. We tested the efficiency of in vivo assembly of a functional nourseothricin-resistance (Nat$^R$) gene from three overlapping PCR products encoding a transcription promoter, protein open reading frame (ORF) and transcription terminator at a single locus using *cis*-multiplexing in the diploid S288C strain. The efficiency of Cas9-mediated integration and assembly of these three DNA fragments to the correct (*URA3*) locus was measured by a combination of 5-fluoroorotic acid resistance (5-FOA$^R$) and Nat$^R$. We found 85% efficiency of targeting and assembly in both copies of the *URA3* locus in diploid yeast S288C cells, and 70% in polyploid ATCC4124 by using the tRNA$^{Pro}$ sequence as the sgRNA promoter (*Figure 3C*; *Supplementary file 1A*), whereas $P_{SNR52}$-dependent targeting was weaker in S288C cells and non-existent in ATCC4124 cells (*Figure 3C*; *Supplementary file 1A*). Thus, CRISPRm enables the one-step markerless assembly of functional genes from multiple fragments into the *S. cerevisiae* genome, resulting in homozygous insertions into both diploid and polyploid yeast strains.

Finally, we tested whether CRISPRm could be used for in vivo selections of improved protein function. We targeted a cellodextrin utilization pathway from the cellulolytic fungus *Neurospora crassa*, comprised of the cellodextrin transporter (*cdt-1*) and an intracellular β-glucosidase (*gh1-1*), to assemble in vivo in diploid *S. cerevisiae*. In combination, these two genes result in yeast cells capable of consuming the disaccharide cellobiose, which could be used to improve biofuel production (*Galazka et al., 2010*). To select for improved cellobiose utilizing strains we used error-prone PCR to amplify the *cdt-1* gene. We co-transformed the library of mutated *cdt-1* alleles, along with overlapping dsDNA fragments for the promoter and terminator, for assembly and integration into the *URA3* locus (*ura3::P_{PGK1}-cdt-1-T_{ADH1}* library) of a diploid strain with a previously integrated β-glucosidase *gh1-1* gene (*lyp1::P_{TDH3}-gh1-1-T_{CYC1}*) using CRISPRm. We then grew the transformants in medium containing cellobiose as the sole carbon source to select for functional *cdt-1* alleles and identified one strain with 2.6-fold (homozygous *cdt-1* N209S/I354N/S360P/T406S/W531L allele) and one strain with 1.7-fold (homozygous *cdt-1* Q45H/F262Y/F533Y allele) increased cellobiose utilization capacity over wild type *cdt-1* (*Figure 4A*). The mutations were mapped in CDT-1 onto structures of the homologous

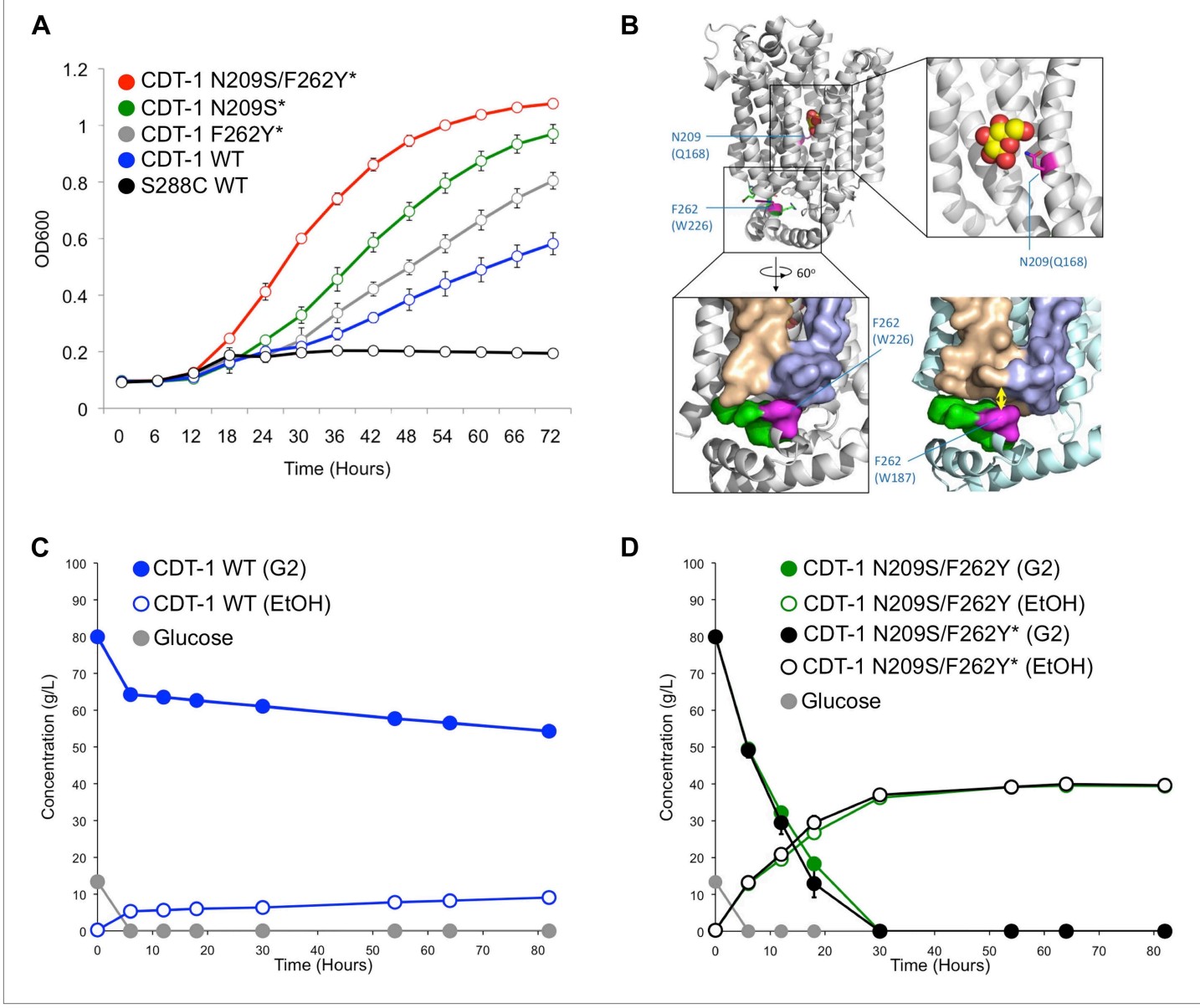

**Figure 4**. CRISPRm mediated insertion and evolution of chromosomal DNA libraries for in vivo protein engineering. (**A**). Utilization of cellobiose in CRISPRm-engineered diploid yeast strains. S288C wild type *cdt-1⁻* (black), *cdt-1* (blue), *cdt-1^{Q45H/F262Y/F533Y}* (indicated as *cdt-1^{F262Y*}* on the figure, gray), *cdt-1^{N209S/I354N/S360P/T406S/W531L}* (indicated as *cdt-1^{N209S*}* on the figure, green) and *cdt-1^{N209S+Q45H/F262Y/F533Y}* (indicated as *cdt-1^{N209S+F262Y*}* on the figure, red). (**B**) Location of mutations at conserved sites in the evolved CDT-1 transporter structure. The mutated residues are colored magenta in the top figure, and are mapped onto the *E. coli* XylE transporter in the outward-facing configuration bound to glucose (upper panels and lower left, PDB entry 4GBZ), and onto the *S. epidermidis* GlcP transporter in the inward-facing configuration (lower right, PDB entry 4LDS). Amino acids in parentheses are the sequences in XylE or GlcP. (**C**) Fermentation of cellobiose by wild-type *cdt-1* expressed from a chromosomally-integrated copy in diploid S288C yeast. Cellobiose is indicated as G2, ethanol as EtOH. The rate of cellobiose consumption was 0.13 g L⁻¹ hr⁻¹. (**D**) Fermentation of cellobiose by mutant versions of *cdt-1* expressed from chromosomally-integrated copies. Glucose consumption (gray) was identical in both strains. The rate of cellobiose consumption for both strains was over 2.0 g L⁻¹ hr⁻¹. In (**C** and **D**), values and error bars represent the means and standard deviations of three independent biological replicates.

major facilitator superfamily (MFS) hexose-like transporters XylE from *Escherichia coli* (**Sun et al., 2012**) and GlcP from *Staphylococcus epidermidis* (**Iancu et al., 2013**) (**Figure 4B**). Notably, one site maps to a site that likely interacts directly with the sugar substrate (N209S) (**Figure 4B**). Whereas two mutations in the second allele (Q45H, F533Y) map to non-conserved regions of the transporters, one

(F262Y) maps to a site immediately adjacent to the nearly universally-conserved PESPR motif in hexose transporters that is involved in transporter dynamics (*Sun et al., 2012*). Furthermore, the mutation from phenylalanine to tyrosine results in an expanded motif (PESPRY) that is present in all of the major hexose transporters in *S. cerevisiae* (*Figure 4B*). We then used CRISPRm to introduce the N209S mutation (nucleotide G626A) in the *cdt-1*$^{Q45H/F262Y/F533Y}$ diploid yeast background. The quadruple mutant *cdt-1* allele resulted in synergistic cellobiose utilization with 3.9-fold increased growth over strains expressing wild-type *cdt-1* (*Figure 4A*). We next tested the selected *cdt-1* alleles in anaerobic fermentations using cellobiose as the sole carbon source. We constructed a *cdt-1*$^{N209S/F262Y}$ double-mutant in diploid yeast using CRISPRm, and compared it to wild-type *cdt-1* and the *cdt-1*$^{Q45H/N209S/F262Y/F533Y}$ quadruple mutant. We found that wild-type *cdt-1* integrated into the chromosome was barely able to support cellobiose fermentation (*Figure 4C*). By contrast, the *cdt-1*$^{Q45H/N209S/F262Y/F533Y}$ quadruple mutant or the *cdt-1*$^{N209S/F262Y}$ double-mutant expressed from the chromosome resulted in complete fermentation of cellobiose, comparable to wild-type *cdt-1* expressed from a high-copy 2μ plasmid (*Figure 4C*; *Ha et al., 2011*).

## Discussion

We have demonstrated that CRISPRm can be used to quickly generate markerless loss-of-function alleles, heterologous insertion alleles, allele swaps and engineer proteins in yeast by in vivo selections for greatly improved metabolic activity. Although we selected for improved protein function, as suggested by the location of the mutations in CDT-1 (*Figure 4B*), it is also possible that selection enriched for sequences that confer improved mRNA stability or translation efficiency, due to the fact that the selection was carried out in vivo. Future experiments will be needed to distinguish between these possibilities.

CRISPRm could be used to integrate DNA libraries to interrogate *cis*-regulatory elements such as transcription promoters, 5′- and 3′-untranslated regions (5′-UTRs and 3′-UTRs) in mRNAs, or riboswitches. CRISPRm enables these types of experiments because it allows for near quantitative integration of DNA libraries, which can be a single fragment or complex assemblies of multiple fragments. This could enable integration and selection of multi-gene pathways, or specific mutation and selection of subsets of a protein or multiple DNA-encoded functions.

The ability to use chromosomally-integrated DNA libraries removes a key limitation in in vivo selection experiments by eliminating biomolecule expression variation due to variable plasmid copy numbers. Although we used a relatively small library size in the present experiments, even in this small library we observed impressive gains in transporter function. We observed 2–3 mutations per kilobase in the selected transporter genes, resulting in 3–5 amino acid substitutions per transporter. We were then able to combine the mutations in a single transporter with superior fermentation performance. Library sizes introduced by CRISPRm should be scalable to the limits of transformation efficiency, to $10^7$ or larger (*Kawai et al., 2010*). CRISPRm should also enable multiple rounds of selection for directed evolution experiments. We envision that each cycle would involve error-prone PCR of the gene of interest, either from a starting sequence or an enriched and improved cell population, followed by CRISPRm-mediated integration of the amplified gene and selection. This cycle would compare favorably with that required for plasmid-based selections, yet would avoid the problem of plasmid copy number variation. CRISPRm should also be adaptable to screens that rely on sorting large populations of cells, that is by fluorescence-activated cell sorting. We envision that CRISPRm could improve the discovery of human therapeutics by yeast display (*Krugel et al., 1988*), and could be used to map the importance of protein–protein interactions (*Ovchinnikov et al., 2014*) via multiplexed library insertion.

To achieve high efficiency multiplexing with CRISPRm, we found it was necessary to modify the sgRNA design by including a 5′ HDV ribozyme, and to assay multiple RNA polymerase III promoters for a given strain background. The ease of discovering tRNAs in any genome and the universality of the sgRNA construct means that, with few modifications, CRISPRm can be used directly in non-model fungal hosts, such as pathogens or organisms used in the biotechnology industry. The genetics of many of these organisms have not been studied in any depth due to the technological limitations of available genetic manipulation techniques. For example, industrial *S. cerevisiae* strains are more stress tolerant and produce much higher yields of desired biofuel or renewable chemical end products than laboratory strains (*Kerr and Service, 2005*; *Farrell et al., 2006*; *Rubin, 2008*). However, linking genotypes of industrial yeasts to their phenotypes remains difficult. CRISPRm should serve as a rapid

and high throughput means for connecting the genotypes of these organisms to their phenotypes, that is for generating marker-free barcoded alleles for large-scale pooled fitness studies of loss-of-function mutants in these organisms (*Giaever et al., 2002*). In the future, CRISPRm may also be applicable to interrogating mammalian cells, if the levels of HDR are sufficiently high. Finally, the ability to use CRISPRm for multiplexed targeting paves the way for applying directed evolution to cellular pathways and genetic circuits for higher order synthetic biology applications in any host strain.

## Materials and methods

### Cloning the pCAS plasmid backbone

Gibson Assembly Mastermix (E2611L) (New England Biolabs, Ipswich, MA) (*Gibson et al., 2009*) was used to fuse the KANMX (Yeastdeletionpages.com) cassette to the pUC bacterial origin of replication from pESC-URA (Agilent Technologies, Santa Clara, CA). Restriction—free (RF) (*van den Ent and Löwe, 2006*) cloning was used to add a yeast 2μ origin of replication from pESC-URA to the pCAS backbone. The resulting pCAS backbone plasmid was propagated in yeast to confirm functionality.

### Cas9 expression constructs

The Cas9 gene from *S. pyogenes* was amplified from clone MJ824 (*Jinek et al., 2012*) and cloned into the pCAS backbone plasmid by RF cloning. A yeast nuclear localization signal (NLS) sequence, codon optimized using IDT software (Integrated DNA Technologies, Coralville, IO), was then cloned into the plasmid by RF cloning. Additional elements fused by RF cloning to the Cas9-NLS sequence included the GFP gene, the CYC1 terminator from *S. cerevisiae* strain S288c (Yeastgenome.org) and the promoters (800 base pairs upstream) from the genes *TDH3*, *TEF1*, *RNR2* and *REV1*, also taken from strain S288c (*Lee et al., 2013*). For genome editing experiments, the GFP sequence was removed from the Cas9 gene and replaced with a C-terminal $His_8$ affinity tag, by RF cloning.

### Engineering of sgRNA constructs

Synthetic DNA (Integrated DNA Technologies, Coralville, IO) for the sgRNA and for a catalytically active form of the Hepatitis Delta Virus (HDV) Ribozyme was sequentially cloned by RF cloning into the Cas9 containing vector (pCAS). The terminator (200 bp) of *SNR52* (Yeastgenome.org) was cloned 3′ of the ribozyme-sgRNA sequence by RF cloning. A series of RNA polymerase III (Pol III) promoters were PCR amplified from S288c genomic DNA and cloned 5′ of the ribozyme-sgRNA sequence by RF cloning. The tRNA promoters included the full-length tRNA plus 100 base pairs upstream of the tRNA gene (*Supplementary file 1C*). The sgRNAs used for multiplex targeting were PCR amplified using primers containing 5′ and 3′ restriction sequences and sub-cloned into pCAS by ligation dependent cloning into SalI, SpeI and SacII unique restriction sites.

### Yeast strains used in this study

*S. cerevisiae* strain S288c (204508; ATCC) (American Type Culture Collection, Manassas, VA) was used as the haploid, and then mated to form the homozygous diploid for the diploid experiments. Yeast strain ATCC4124 is an industrial polyploid yeast isolated from a molasses distillery, and was obtained from ATCC (American Type Culture Collection, Manassas, VA).

### Cas9-GFP localization and expression

Expression and localization of Cas9-GFP was verified by imaging haploid *S. cerevisiae* S288c cells transformed with pCas9-GFP::KAN using fluorescence microscopy (Leica Epifluorescence, Leica Microsystems, Buffalo Grove, IL). Cells were grown overnight and nuclear localization visualized at 100× magnification.

### Fitness analysis of Cas9 expressed by different promoters

Yeast cells containing pCAS (Cas9-$His_8$ variant) were grown in a Bioscreen C Growth Curve Analyzer (Growth Curve USA, Piscataway, NJ) in 200 μl of YPD + G418 liquid medium (20 g/l Peptone [211667; Bacto], 10 g/l Yeast Extract [212750; Bacto], 0.15 g/l Adenine hemisulfate [A9126; Sigma] and 20 g/l Glucose [G8270; Sigma] + 200 mg/l G418 [29065A; Santa Cruz Biotechnology]). Cells were grown in three biological replicates each with five technical replicates for 48 hr at 30°C under constant shaking. The wild-type control containing an empty vector (pOR1.1) was also grown in five technical replicates. Mean and standard deviations of the optical density at 600 nm were calculated for each time point measured by the Bioscreen.

## CRISPR-Cas9 screening protocol

The Cas9 transformation mix consisted of 90 µl yeast competent cell mix (OD$_{600}$ = 1.0), 10.0 µl × 10 mg/ml ssDNA (D9156; Sigma, St. Louis, MO), 1.0 µg pCAS plasmid, 5.0 µg of linear repair DNA and 900 µl Polyethyleneglycol$_{2000}$ (295906; Sigma), 0.1 M Lithium acetate (517992; Sigma) 0.05 M Tris–HCl (155568; Invitrogen) EDTA (10618973; Fisher Scientific). To measure Cas9 independent integration, the linear DNA was co-transformed with a plasmid lacking the Cas9 protein and sgRNA (pOR1.1) (*Supplementary file 2B*). Cells were incubated 30 min at 30°C, and then subjected to heat shock at 42°C for 17 min. Following heat shock, cells were re-suspended in 250 µl YPD at 30°C for 2 hr and then the entire contents were plated onto YPD+G418 plates (20 g/l Peptone, 10 g/l Yeast Extract, 20 g/l Agar, 0.15 g/l Adenine hemisulfate, 20 g/l Glucose and G418 at 200 mg/l). Cells were grown for 48 hr at 37°C, and colonies imaged using the Biorad ChemiDoc Imager (Biorad, Hercules, CA) before replica plating onto phenotype-selective media. The guide sequences in the sgRNAs used for targeting the various loci are shown in *Supplementary file 1D*. *URA3* mutants were selected on 2.0 g/l Yeast nitrogen base without amino acids or ammonium sulfate (Y1251; Sigma), 5.0 g/l Ammonium sulfate (A4418; Sigma), 1.0 g/l CSM (4500-012; MP Biosciences), 20 g/l Glucose, 20 g/l Agar + 5-fluoroorotic acid (1 g/l) (F-230-25; Goldbio); *LYP1* mutants were selected on 2.0 g/l Yeast nitrogen base without amino acids or ammonium sulfate, 5.0 g/l Ammonium sulfate, 1.0 g/l CSM-lysine (4510-612; MP Biosciences), 20 g/l Glucose, 20 g/l Agar + thialysine (100 mg/l) (A2636; Sigma); *CAN1* mutants were selected on 2.0 g/l Yeast nitrogen base without amino acids or ammonium sulfate, 5.0 g/l Ammonium sulfate, 1.0 g/l CSM-arginine (4510-112; MP Biosciences), 20 g/l Glucose, 20 g/l Agar + canavanine sulfate (50 mg/l) (C9758; Sigma); the remaining auxotrophic mutants were selected on 2.0 g/l Yeast nitrogen base without amino acids or ammonium sulfate, 5.0 g/l Ammonium sulfate, 1.0 g/l CSM , 20 g/l Glucose, 20 g/l Agar; and aerobic respiration deficient mutants (*petites*) were selected on 20 g/l Peptone, 10 g/l Yeast Extract, 20 g/l Agar, 0.15 g/l Adenine hemisulfate, 20 g/l Glycerol (G5516; Sigma). Colonies from the YPD+G418 plates were picked and grown overnight in 0.8 ml of YPD. Genomic DNA was extracted from these cultures using the MasterPure Yeast DNA Extraction Kit (MPY80200; Epicentre). PCR confirmation of the 60-mer integration allele was performed using primers flanking the target site. PCR products were purified by Exo-SAP-IT (78201; Affymetrix) and Sanger sequenced at the UC Berkeley, Sequencing Facility (Berkeley, CA) to confirm barcode sequence in the amplicon.

## RT-qPCR of sgRNAs

Cells containing the pCAS plasmid with sgRNA inserts were grown in 900 µl of YPD+G418 medium for 24 hr at 30°C and 750 rpm. Total RNA was extracted from exponentially growing yeast cells using Ambion RNA RiboPure Yeast Kit (AM1926) (Life Technologies, Carlsbad, CA). RT-qPCR was performed on the Applied Biosciences StepOne Real-Time PCR System (Applied Biosystems, Foster City, CA) using the Invitrogen EXPRESS One-Step SYBR GreenER Kit (Life Technologies, Carlsbad, CA). The RT-qPCR expression level data was quantified using the Comparative CT$_T$ ($\Delta\Delta C_T$) method and relative abundance of the sgRNA was normalized to the mRNA transcript *UBC6*, which was used as the endogenous control (*Teste et al., 2009*). The primer sequence used for the RT reaction was 5'-AAAAGCACCGACTCGGT-3' and the additional q-PCR primer used was 5'-GTTTTAGAGCTAGAAATAGCAAG-3'. The primers used for the *UBC6* endogenous control were (RT) 5'-CATTTCATAAAAAGGCCAACC-3' and (qPCR) 5'-CCTAATGATAGTTCTTCAATGG-3'. DNA sequences for the sgRNAs used to test HDV ribozyme function are shown in *Supplementary file 2C* and *Supplementary file 2D*.

## Multiplex genome targeting by Cas9

Multiplex targeting was performed as described above using pCAS plasmids containing more than one sgRNA expression construct cloned into one of the restriction sites by ligation dependent cloning. Single vs double mutant efficiency was scored relative to the number of colonies present on the YPD+G418 plate. Genomic DNA isolation and PCR of the integration site was performed as described above.

## Multiplex in vivo assembly of DNA using Cas9

Drug resistance cassettes were assembled in vivo from three linear double-stranded DNA fragments PCR amplified from the *Ashbya gossypii TEF1* promoter (*AgP$_{TEF1}$*), the nourseothricin open reading frame (Nat$^R$) and *A. gossypii TEF1* (*AgT$_{TEF1}$*) terminator in separate reactions. The primers used to

amplify the promoter and terminators contained 50 bp of homology to the nourseothricin ORF and 50 bp of homology to the genomic target. Colonies exhibiting drug resistance (nourseothricin 100 mg/l) (N-500-1; Goldbio) following replica plating were compared to the number of colonies on the YPD+G418 to determine efficiency of multiplex assembly.

## Selection of improved variants of cellodextrin transporter CDT1

To generate *CDT1* mutant allele libraries, the GeneMorph II Random Mutagenesis Kit (200550; Aglient) (Agilent Technologies, Santa Clara, CA) was used to amplify the *N. crassa cdt-1* open reading frame. The library of *cdt-1* mutant alleles was co-transformed with PCR-amplified linear dsDNA fragments encoding the *ScP*$_{PGK1}$ promoter and *ScT*$_{ADH1}$ terminator into a diploid S288c yeast strain containing a previously-integrated *gh1-1* gene. The *gh1-1* gene included the *ScP*$_{TDH3}$ promoter, the *N. crassa gh1-1* open reading frame and *ScT*$_{CYC1}$ terminator. The primers used to amplify the promoters and terminators contained 50 bp of homology to either the *cdt-1 or gh1-1* ORFs and 50 bp of homology to the respective the genomic targets. 5 µg of each of the three PCR products (promoter, open reading frame, terminator) were co-transformed with the pCAS plasmid containing the *URA3* guide sequences and screened for G418 resistance as described. Approximately 1600 G418-resistant colonies were pooled and resuspended in minimal cellobiose medium (SCel) (2.0 g/l Yeast nitrogen base without amino acids or ammonium sulfate, 5.0 g/l Ammonium sulfate, 1.0 g/l CSM, 20 g/l Cellobiose). Resuspended cells were immediately spread evenly on SCel plates for initial analysis prior to cellobiose selection (*t* = 0 samples). Ten microliters of the resuspended cells were inoculated in 50 ml of SC medium in biological triplicate. Cells were harvested after 3 days and spread onto SCel plates. Cells were grown at 30°C for 4 days, and the largest colonies were chosen for further analysis.

## Tecan growth analyzer and fitness calculation of cdt-1 alleles

Cells were grown overnight in 0.5 ml of Synthetic Dextrose (2%) (SD) in 96 well plates. Cultures were diluted 1:500 in SCel media (4% cellobiose) and 150 µl were grown using the Tecan Sunrise (Tecan Systems Inc., San Jose, CA) in biological triplicate for 3 days at 30°C. Average and standard deviation was calculated for each biological sample. Relative fitness was calculated by measuring area between the curves (ABC) which normalizes growth to the area under the curve (AUC) for diploid cells lacking *cdt-1* (ABC = AUC *cdt-1*$^{+}$−AUC *cdt-1*$^{-}$). Fold cellobiose utilization capacity = (ABC *cdt-1*$^{S209}$/ABC *cdt-1*).

## Generating the double mutant and quadruple cdt-1 alleles

Integration of the repair oligonucleotide into the chromosomal wild-type *cdt-1* gene or *cdt-1*$^{Q45H/F262Y/F533Y}$ was performed as described for integrating barcoded DNA. The sgRNA sequence used to target the wild type allele of *cdt-1* was cloned into the pCAS vector with the sequence 5′-TGCACTGGC TTCTACAACTG-3′. The repair oligonucleotide used to make the A626G (i.e., N209S) mutation was 5′-CGGCCGCTGCACTGGCTTCTACAGCTGCGGTTGGTTCGGAGGTTCATTCCTGCCGCCTG-3′ with 50 more base pairs of homology to *cdt-1*, on both sides of the above oligonucleotide. Genomic DNA was extracted as described and Sanger sequencing confirmed the incorporation of the A626G mutation. Similarly, the T785A mutation (F262Y) was incorporated into the *cdt-1*$^{N209S}$ allele using an sgRNA with guide sequence 5′-CCTCGCTTCCTATTTGCCAA-3′, and a repair oligonucleotide with sequence 5′-AGAATCCCCTCGCTACCTATTTGCCAACGGCCGCGACGCTGAGGCTGTTGCCTTTCTTGT-3′. Genomic DNA was extracted as described and Sanger sequencing confirmed the incorporation of the T785A mutation.

### Structural analysis of cdt-1$^{N209S/Q45H/F262Y/F533Y}$

The sequences of *N. crassa* CDT-1 (accession XP_963801), *E. coli* XlyE (accession YP_492174.1), and *S. epidermidis* GlcP glucose transporter (accession ZP_04818045.1) were aligned with the MUSCLE algorithm (*Edgar, 2004*). Figures were prepared with the PyMol molecular graphics system (http://www.pymol.org/).

### Imaging of integrated and plasmid based fluorescent proteins

Strain yML068 was generated by integrating a short DNA 'spacer' sequence and Leu2 into the *LEU2* locus of BY4741. Strain yML069 was generated by integrating *P*$_{TDH3}$-mRuby2-*T*$_{TDH1}$ and Ura3 into the *URA3* locus, and integrating *P*$_{TDH3}$-Venus-*T*$_{TDH1}$ and Leu2 into the *LEU2* locus of BY4741. Strain yML104 was generated by integrating *P*$_{TDH3}$-mRuby2-*T*$_{TDH1}$-*P*$_{TDH3}$-Venus-*T*$_{TDH1}$ and Ura3 into the *URA3* locus of

yML068. Plasmid pML1350 is a Ura3 CEN6/ARS4 plasmid with $P_{TDH3}$-mRuby2-$T_{TDH1}$. pML1362 is a Leu2 CEN6/ARS4 plasmid with $P_{TDH3}$-Venus-$T_{TDH1}$. pML1177 is a Ura3 CEN6/ARS4 plasmid with $P_{TDH3}$-mRuby2-$T_{TDH1}$-$P_{TDH3}$-Venus-$T_{TDH1}$.

Synthetic defined media lacking leucine and uracil (SD-LU) was made by adding 6.7 g/l Difco Yeast Nitrogen Base without amino acids; 2 g/l Drop-out Mix Synthetic Minus Leucine, Uracil without Yeast Nitrogen Base (US Biological); and 20 g/l Dextrose to distilled water.

Colonies were picked into SD-LU media and grown to exponential phase at 30°C. Cultures were concentrated by centrifugation, spotted onto plain glass slides, and examined on a Zeiss Observer D1 microscope using a 100× DIC objective. Images were captured using a Hamamatsu Orca-flash 4.0 (C11440) camera using auto-exposure. Fluorescence images were taken using an X-Cite Series 120 lamp, Zeiss filter set 45 (excitation at 560/40 nm and emission at 630/75 nm) for mRuby2, and Zeiss filter set 46 (excitation at 500/20 nm and emission at 535/30 nm) for Venus. Images were analyzed and composites were created using Fiji (http://fiji.sc).

## Illumina sequencing to identify potential off-target mutations

To rule out that our CRISPR-Cas9/sgRNA system resulted in non-specific genome targeting, we performed whole genome sequencing for *URA3* and *LYP1* targeted strains and searched for insertion/deletions (INDELs), single nucleotide polymorphisms (SNPs) and multi-nucleotide polymorphisms (MNPs). We identified 21 sequence variants across the nine *URA3* and *LYP1* targeted strains (*Supplementary file 1B*). Whole genome sequencing was performed by the UC Davis Genome Center (Davis, CA) using the Illumina MiSeq platform (Illumina, Hayward, CA) to produce 150 bp paired-end reads. The software package versions used for sequencing data analysis were as follows: BWA (v. 0.7.5ar405), Picard (v. 1.92[1464]), SAMtools (v. 0.1.19-44428cd) and the GATK (2.7-2-g6bda569). The S288C reference genome (v. R64-1-1, release date 3 Feb 2011) was obtained from the *Saccharomyces* Genome Database (yeastgenome.org) and prepared for use in sequencing data analysis with bwa index, CreateSequenceDictionary from Picard, and samtools faidx. Sequencing reads were processed with Scythe (v. 0.991) to remove adapter contamination and Sickle (v. 1.210) to trim low quality bases. Processed reads were mapped to the S288C reference genome using bwa mem with the—M option for picard and GATK compatibility. The mapped reads were sorted with SortSam and duplicate reads were marked with MarkDuplicates from Picard. Read alignments were refined by performing local realignment with the RealignerTargetCreator and IndelRealigner walkers from the GATK on all samples collectively.

Variant detection for both SNPs and INDELs was performed with GATK's UnifiedGenotyper, with parameters adjusted for haploid genomes and no downsampling of coverage, for each sample independently. The resulting SNP and INDEL calls were filtered with the VariantFiltration walker from GATK (see header of the VCF file, supplemental VCF file, for details). A custom Perl script (*Supplementary file 3*) was written to identify all GG dinucleotide sequences in the S288C reference genome, extract every Cas9 target sequence (i.e., 20 nt sequence corresponding to the 20 nucleotides immediately 5′ of the 'NGG' PAM site), and obtain the genome coordinates ranging from 30 nucleotides upstream and downstream of the PAM site. Cas9 target sequences were added to VCF files as custom annotations using snpEff (v3.3h), and SnpSift (v3.3h) was used to extract desired fields into tables for analysis with a custom R script (*Supplementary file 4*). Needleman–Wunsch global alignments between our guide sequences and Cas9 target sequences were performed using the pairwiseAlignment function (Biostrings package, Bioconductor) in R, with a substitution matrix of −1 for mismatches and 2 for matches, produced with the nucleotideSubstitutionMatrix function (Biostrings package, Bioconductor). The probability of there being a better match for the guide sequence to a given Cas9 target sequence was calculated as the frequency of Cas9 target sequences with better alignments to the same guide sequence, amongst 10,000 randomly selected Cas9 target sequences. To compile counts of all variants and various subclasses, a GATKReport was generated from the VCF files with GATK's VariantEval walker, read into R using the 'gsalib' library, and the desired categories were extracted with a custom R script (*Supplementary file 4*).

Without the repair DNA template required for HDR, the majority of mutations caused by Cas9 are expected to be INDELs, SNPs or MNPs that initiate within the protospacer sequence, which is the 20 nucleotide (nt) sequence 5′ of the PAM (*Fu et al., 2013*). Therefore, for completeness, we searched for all PAM sites within 30 nt upstream and downstream for each of the 21 variants. We then compared the 20 nt *URA3* and *LYP1* guide sequences to the putative protospacer sequences within the regions

flanking each variant. An end-to-end alignment identified 10 or fewer nucleotide matches between the *URA3* or *LYP1* guide sequences and the variant sequences (**Supplementary file 1B**). Cas9 requires at least 12 perfect base pair matches within the guide-target sequence (**Hsu et al., 2013**) so it is highly unlikely that the *URA3* and *LYP1* guide sequences directed Cas9 to any of these potential protospacers that lie within variant sites. These mutations likely arose as natural variants during the course of the experiment.

As a second method to evaluate the likelihood of off-target mutations, we performed local alignments of our guide sequences to all Cas9 target sequences whose PAM site was within 30 nucleotides upstream and downstream of a detected variant, as well as to 10,000 randomly selected Cas9 target sequences from the genome. Since our guide sequences are expected to have a better match to 13% or more of all Cas9 target sequences (~126,000 or more sites) than to the best matching Cas9 target sequence with a nearby variant, and the number of nucleotide matches in end-to-end alignments is at most 10, we argue that the variants identified in the genomes of *URA3*- and *LYP1*-targeted strains are unlikely the result of off-target Cas9 modifications.

## Anaerobic fermentations

Yeast strain colonies were inoculated in 20 ml of oMM (optimized minimal media) (Yuping Lin, personal communication) with 2% glucose in 50 ml Falcon tubes and grown aerobically at 30°C to saturation for 24 hr. The oMM contained 1.7 g/l YNB (Y1251; Sigma, Saint Louis, MO, USA), 2× CSM, 10 g/l $(NH_4)_2SO_4$, 1 g/l $MgSO_4 \cdot 7H_2O$, 6 g/l $KH_2PO_4$, 100 mg/l adenine hemisulfate, 10 mg/l inositol, 100 mg/l glutamic acid, 20 mg/l lysine, 375 mg/l serine and 0.1 M 2-(*N*-morpholino) ethanesulfonic acid (MES) pH 6.0. The saturated cultures were then inoculated to starting OD600 of 0.2 in 500 ml of oMM with 2% glucose in 1 l Erlenmeyer flasks and grown aerobically at 30°C to mid-log phase at a final OD600 of 2.5. Cells were harvested and washed twice with sterile $ddH_2O$. The washed cells were then inoculated at OD600 of 20 in 50 ml oMM with 1% glucose and 8% cellobiose in 125 ml serum flasks. After inoculation, the flasks were sealed with rubber stoppers and clamped with an aluminum seal. To achieve anaerobic conditions, the headspaces of the sealed flasks were purged with nitrogen gas for 30 min. These were then cultivated at 30°C and 220 rpm.

Using sterile needles and syringes, 1 ml samples were removed through the rubber stoppers at the indicated time points. The cells were pelleted and 5 μl of the supernatants were analyzed for cellobiose, glucose, glycerol, and ethanol content by high performance liquid chromatography on a Prominence HPLC (Shimadzu, Kyoto, Japan) equipped with Rezex RFQ-FastAcid H 10 × 7.8 mm column. The column was eluted with 0.01 N of $H_2SO_4$ at a flow rate of 1 ml/min, 55°C.

## Acknowledgements

We thank D Nunn and J Doudna for helpful discussions. We thank J Waters for assistance in phenotype screening assays. This work was supported by funding from the Energy Biosciences Institute.

## Additional information

### Competing interests

OWR: A patent application related to this work has been filed by J Cate and O Ryan on behalf of the Regents of the University of California. JHDC: A patent application related to this work has been filed by J Cate and O Ryan on behalf of the Regents of the University of California. The other authors declare that no competing interests exist.

### Funding

| Funder | Author |
| --- | --- |
| Energy Biosciences Institute | Owen W Ryan, Jeffrey M Skerker, Matthew J Maurer, Xin Li, Jordan C Tsai, Snigdha Poddar, Michael E Lee, Will DeLoache, John E Dueber, Adam P Arkin, Jamie HD Cate |

The funder had no role in study design, data collection and interpretation, or the decision to submit the work for publication.

## Author contributions

OWR, JMS, MEL, WDL, Conception and design, Acquisition of data, Analysis and interpretation of data, Drafting or revising the article; MJM, XL, JCT, SP, Acquisition of data, Analysis and interpretation of data, Drafting or revising the article; JED, APA, JHDC, Conception and design, Analysis and interpretation of data, Drafting or revising the article

## Additional files

### Supplementary files

• Supplementary file 1. (**A**) CRISPR-Cas screening results. (**B**) Potential off-target mutations identified by whole genome sequencing. (**C**) RNA polymerase III promoter sequences used to express sgRNAs. (**D**) Guide sequences used in this study.

• Supplementary file 2. (**A**) pCAS—Tyrosine—sgRNA-LYP1. (**B**) pOR1.1. (**C**) sgRNA without HDV ribozyme. (**D**) sgRNA with HDV ribozyme.

• Supplementary file 3. Custom perl script for Cas9 target identification in S288C genome.

• Supplementary file 4. Custom R script for yeast genome analysis.

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
