## [Decision Letter]

Thank you for sending your work entitled “Evolution of chromosomal DNA libraries using CRISPRm” for consideration at *eLife*. Your article has been favorably evaluated by Richard Losick (Senior editor) and 2 reviewers, one of whom is a member of our Board of Reviewing Editors.

The Reviewing editor and the other reviewers discussed their comments before we reached this decision, and the Reviewing editor has assembled the following comments to help you prepare a revised submission.

The reviewers agree that the manuscript presents very important technological improvements that will be of interest to the yeast community and beyond. The strong point of the manuscript is the systematic comparison of different parameters to optimize the use of the CRISPR-Cas9 system in different yeast strains. This will be of wide interest to researchers in the yeast community and beyond to those working in genome engineering and synthetic biology.

However, the “protein evolution” experiment presented to validate the methodological development was judged to be of limited interest. The major concerns raised by these experiments are listed below. Although we do not request additional experiments to be performed, we would like to ask you to discuss these points in the revised version of manuscript.

Major concerns:

1) In the “protein evolution” experiment, a single cycle of mutation was done and it is not clear how more complex selection procedures would be implemented.

2) If mutated coding sequences conferring improved characteristics to the host strain were selected, the characterization of the cognate proteins has not been performed. Thus at this stage, it remains unclear whether protein activity, stability, localization or even mRNA translatability of stability was changed.

3) Also, the mutation rate resulting from the mutagenic PCR is unknown.

4) The screening of complex libraries will certainly require screening higher numbers of transformants by several orders of magnitude. (Only 1600 transformants screened here, is it meaningful?)

5) How will screening of very large numbers of transformants be achieved?

---

## [Author Response]

We have modified the manuscript to address the reviewers’ comments, which we think has improved the readability of the paper. We also include 2 custom scripts (one Perl, one R) that we used to analyze the yeast genome sequences ([Supplementary-material SD3-data SD4-data]). We also edited those parts of the Methods section to clarify what the scripts do.

*The reviewers agree that the manuscript presents very important technological improvements that will be of interest to the yeast community and beyond. The strong point of the manuscript is the systematic comparison of different parameters to optimize the use of the CRISPR-Cas9 system in different yeast strains. This will be of wide interest to researchers in the yeast community and beyond to those working in genome engineering and synthetic biology*.

*However, the “protein evolution” experiment presented to validate the methodological development was judged to be of limited interest. The major concerns raised by these experiments are listed below. Although we do not request additional experiments to be performed, we would like to ask you to discuss these points in the revised version of manuscript*.

We thank the reviewers for their positive evaluation of our work. We will describe in more detail our changes to the main text below. We note here that we have changed the title in line with suggestions from the reviewers and *eLife* editors to be, “Selection of chromosomal DNA libraries using a multiplex CRISPR system.”

*1) In the “protein evolution” experiment, a single cycle of mutation was done and it is not clear how more complex selection procedures would be implemented*.

We think there are many variations that could be envisioned. In the case of using error-prone PCR, subsequent iterations of selection would involve PCR amplification of the gene from the enriched cell population using error-prone conditions, followed by a new round of integration and selection. Each cycle would thus be:

A) Error-prone PCR amplification (either from starting plasmid or genomic DNA from cells of increased fitness isolated in the previous round).

B) CRISPRm-mediated integration of the amplicons.

C) Selection.

We have incorporated the envisioned cycle into the Discussion section, as described in our answer to major point 4, below.

*2) If mutated coding sequences conferring improved characteristics to the host strain were selected, the characterization of the cognate proteins has not been performed. Thus at this stage, it remains unclear whether protein activity, stability, localization or even mRNA translatability of stability was changed*.

We agree that the underlying mechanism remains to be determined. We have added a statement to this effect, as a goal for future experiments, in the first paragraph of the Discussion:

“Although we selected for improved protein function, as suggested by the location of the mutations in CDT-1 (Figure 4), it is also possible that selection enriched for sequences that confer improved mRNA stability or translation efficiency, due to the fact that the selection was carried out *in vivo*. Future experiments will be needed to distinguish between these possibilities.”

*3) Also, the mutation rate resulting from the mutagenic PCR is unknown*.

We agree that the optimal level of mutagenic PCR could be optimized in the future. We have added a statement addressing what we know about the mutation rate in our experiment, in the 3^rd^ paragraph of the Discussion:

We observed 2-3 mutations per kilobase in the selected transporter genes, resulting in 3-5 amino acid substitutions per transporter. We were then able to combine the mutations in a single transporter with superior fermentation performance.

4) The screening of complex libraries will certainly require screening higher numbers of transformants by several orders of magnitude. (Only 1600 transformants screened here, is it meaningful?)

We demonstrated a large improvement with a small library, which we would argue is a positive. Although the library size here was small, we were able to isolate in one experiment a better transporter than the entire field working on this problem had been able to achieve in the past 3-4 years. CRISPRm can be scaled simply by transforming more linear repair DNA. We believe that significant improvement within a single round of selection by using a relatively small library is a powerful proof-of-principle result. We did not want to find too many mutations to follow up as we are in an ultra-competitive CRISPR field. Obviously, increasing the scale and iterations of CRISPRm would yield more evolved proteins until one reaches an optimum. We therefore think that there is only upside potential to the approach with further optimization.

To address this point, we modified the Discussion as follows (see further our response to point 1 above):

“Although we used a relatively small library size in the present experiments, even in this small library we observed impressive gains in transporter function. Library sizes introduced by CRISPRm should be scalable to the limits of transformation efficiency, to 10^7^ or larger (18). CRISPRm should also enable multiple rounds of selection for directed evolution experiments. We envision that each cycle would involve error-prone PCR of the gene of interest, either from a starting sequence or an enriched and improved cell population, followed by CRISPRm-mediated integration of the amplified gene and selection. This cycle would compare favorably with that required for plasmid-based selections, yet would avoid the problem of plasmid copy number variation.”

5) How will screening of very large numbers of transformants be achieved?

The library size is scalable by transforming more linear DNA. We purposely kept it small so that we had a manageable number of improved mutants to follow up. We wanted a small library to demonstrate the efficiency. We note that the actual selection for improved transporters was carried out in small volumes of liquid media, which we think could easily be scaled up.